# Absent thermal equilibration on fractional quantum Hall edges over macroscopic scale

Ron Aharon Melcer [1✉], Bivas Dutta [1], Christian Spånslätt [2,3,4], Jinhong Park [5], Alexander D. Mirlin[3,4,6,7] & Vladimir Umansky[1]

Two-dimensional topological insulators, and in particular quantum Hall states, are characterized by an insulating bulk and a conducting edge. Fractional states may host both downstream (dictated by the magnetic field) and upstream propagating edge modes, which leads to complex transport behavior. Here, we combine two measurement techniques, local noise thermometry and thermal conductance, to study thermal properties of states with counter-propagating edge modes. We find that, while charge equilibration between counter-propagating edge modes is very fast, the equilibration of heat is extremely inefficient, leading to an almost ballistic heat transport over macroscopic distances. Moreover, we observe an emergent quantization of the heat conductance associated with a strong interaction fixed point of the edge modes. Such understanding of the thermal equilibration on edges with counter-propagating modes is a natural route towards extracting the topological order of the exotic 5/2 state.

[1] Braun Center for Submicron Research, Department of Condensed Matter Physics, Weizmann Institute of Science, Rehovot 761001, Israel. [2] Department of Microtechnology and Nanoscience, Chalmers University of Technology, S-412 96 Göteborg, Sweden. [3] Institute for Quantum Materials and Technologies, Karlsruhe Institute of Technology, 76021 Karlsruhe, Germany. [4] Institut für Theorie der Kondensierten Materie, Karlsruhe Institute of Technology, 76128 Karlsruhe, Germany. [5] Institute for Theoretical Physics, University of Cologne, Zülpicher Str. 77, 50937 Köln, Germany. [6] Petersburg Nuclear Physics Institute, 188300 St, Petersburg, Russia. [7] L. D. Landau Institute for Theoretical Physics RAS, 119334 Moscow, Russia. ✉email: ron.melcer@weizmann.ac.il

The quantum Hall effect (QHE) is perhaps the most studied two-dimensional topological phenomenon. Whereas excitations in the sample bulk are localized, gapless excitations flow in one-dimensional chiral modes along the edge[1]. The integer quantum Hall effect—emerging from quantization of electron cyclotron orbits and an integer occupation of Landau levels—is well understood within a single electron picture. By contrast, the richer fractional quantum Hall effect arises from strong electron–electron interactions. Among fractional states, particularly peculiar are the hole-conjugate states (at fillings $\nu = \frac{p}{2p-1}$, with integer $p > 1$, i.e., 2/3, 3/5, 4/7,…) since they host upstream propagating edge modes. Such modes have been observed for various filling factors and devices[2–5].

Topological properties of the bulk are reflected in the edge structure, allowing edge-transport coefficients to be quantized. In particular, with full charge equilibration (e.g., by impurity scattering) between counter-propagating modes, the two-terminal electrical conductance is given by $G_{2T} = \frac{e^2}{h}\nu$, where $\nu$ is the filling factor. Likewise, for thermally equilibrated edges, the two-terminal heat conductance is quantized[6,7] as $G_{2T}^Q = \kappa_{2T} T = |\nu_Q|\kappa_0 T$, where $\kappa_0 = \pi^2 \frac{k_B^2}{3h}$, with $k_B$ the Boltzmann constant, $T$ the temperature, and $\nu_Q$ is an integer (or a fraction for non-Abelian states[8]). Importantly, the value of $\nu_Q$ is an inherent property of the bulk topological order. Specifically, for Abelian states, $\nu_Q$ is given by the net number of edge modes, $\nu_Q = n_d - n_u$, with $n_d(n_u)$ being the number of downstream (upstream) modes. In hole-conjugate states, $\nu_Q$ can be zero or negative, implying transport of heat in the direction opposite to the charge flow. In recent years, thermal conductance measurements were successfully performed in GaAs and in graphene[9–12], manifesting the quantization of thermal conductance for both integer and thermally equilibrated fractional states. Nonetheless, a detailed understanding of thermal equilibration on the edge is crucial for interpreting[13–17] the recent observation[11] of $\kappa_{2T} \approx 2.5\kappa_0$ at filling $\nu = \frac{5}{2}$. As both the topological order and the extent of thermal equilibration are not known, two contradicting explanations were proposed: (i) full thermal equilibration[11], which implies $\nu_Q = 2.5$, indicating a topological order known as the PH-Pfaffian, which is further supported by a recent experiment[18], (ii) partial thermal equilibration[15] and $\nu_Q = 1.5$, indicating the anti-Pfaffian order, which is supported by numerical simulations[19,20]. Our present work paves the way towards the solution, by combining local thermometry with thermal conductance measurements to study thermal equilibration in Abelian fractional states, for which the topological order is known.

The interplay between topology and equilibration also determines the temperature profile along the edge. Consider an edge with two contacts attached at $x = 0$ and $x = L$, and upstream modes sourced at an elevated temperature $T_u(0) = T_m$, while downstream modes emerge from a cold drain contact (for simplicity assumed to be at zero temperature: $T_d(L) = 0$). With efficient thermal equilibration (thermal equilibration length $l_{eq} \ll L$), the temperature of the upstream modes after propagation, $T_u(L)$, is expected to be qualitatively different for three topologically distinct classes[21] determined by the sign of $\nu_Q$: (i) for $\nu_Q > 0$ (e.g., in $\nu = 5/3$), upstream modes lose energy propagating upstream and arrive cold at the drain up to exponential corrections: $T_u(L) \sim T_m e^{-\frac{L}{l_{eq}}}$, (ii) for $\nu_Q < 0$ ($\nu = 3/5$), the injected heat propagates ballistically to the drain up to exponential corrections: $T_u(L) \sim T_m(\text{const} + e^{-\frac{L}{l_{eq}}})$, (iii) for $\nu_Q = 0$ ($\nu = 2/3$), the thermal transport is diffusive rather than ballistic, resulting in $T_u(L) \sim T_m(\frac{l_{eq}}{L})^{\frac{1}{2}}$.

A competing process along the edge is energy dissipation (loss to the environment). Heat, unlike charge, can escape from edge modes to phonons, photons (due to stray capacitances), or neutral excitations in the bulk (localized states coupled by Coulomb interaction). These processes cause an exponential decay in the upstream temperature, $T_u(L) = T_m e^{-\frac{L}{l_{dis}}}$, with $l_{dis}$ a characteristic dissipation length. Such dissipation is a compelling explanation to recent observations of relaxation of heat flow in particle-like states (with $n_u = 0$)[22]. If the thermal equilibration is weak compared to the energy dissipation ($l_{dis} \ll l_{eq}$), energy backscattering is of no importance, and an exponentially decaying profile of $T_u$ is expected regardless of the state's topology[23].

## Results

**Neutral mode thermometry.** In order to measure the local temperature of upstream modes, we fabricated devices based on a high-mobility two-dimensional electron gas (2DEG), embedded in a GaAs-AlGaAs heterostructure, with density $8.2 \times 10^{10}$ cm$^{-2}$ and mobility $4.4 \times 10^6$ cm$^2$V$^{-1}$s$^{-1}$. Device A consists of three Ohmic contacts: source (S), an upstream located amplifier contact (A), and a ground contact (G) (see Fig. 1). The propagation length $L$, between S and A, could be varied using three metallic gates, which, when negatively charged, force the edge modes to take a detour, thus elongating the propagation length. When bias is applied to the source, power is dissipated at the back of the contact, leading to a hotspot[24] (depicted as a red fire) with an elevated temperature $T_m$.

The temperature of the upstream modes reaching A was determined from current fluctuations measured in A. This upstream noise is a smoking-gun signature of the presence of upstream modes[3,4], as studied theoretically recently[21,25]. The noise is generated in a noise spot (depicted as a white bolt sign in Fig. 1), a region with size of the charge equilibration length outside A. The existence of this noise spot is a consequence of counter-propagating edge modes and efficient charge equilibration[21,25]. The elevated temperature of upstream modes at the noise spot excites particle-hole pairs. If a particle (or hole) is absorbed by the amplifier contact while the hole (or particle)

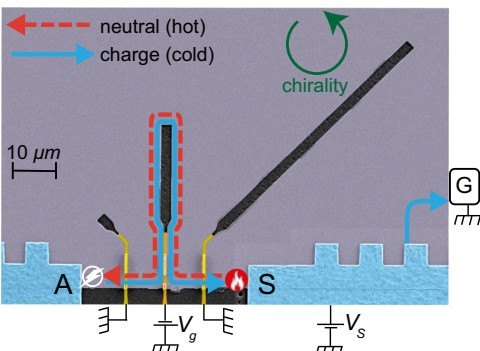

**Fig. 1 Device.** False colors SEM image of the heart of Device A. This device consists of three Ohmic contacts at the edge of the MESA (colored gray): Source (S), Amplifier (A) and the cold-grounded drain (G) (shown by symbol only). The propagation length of the edge modes between the S and A contacts can be tuned by using the three metallic gates (light-yellow: unbiased, dark-yellow: biased), which upon biasing, add the etched regions inside the MESA (funnel shaped black regions) to the upstream path. Applying a voltage $V_s$ on S causes the formation of a hotspot (marked with red fire) at the back of S. The upstream modes (red dashed line) emanating from the hotspot, carry the heat to A where the noise is generated (marked with white bolt). We depict here the path of the edge modes for the situation where the middle gate (dark-yellow) redirects the edge modes with the application of a gate voltage $V_g$, while the other two gates remain unbiased (light-yellow), and hence do not affect the edge modes' path. Zero bias on all gates forms the shortest propagation length (straight line from S to A), while biasing all three gates forms the longest propagation length.

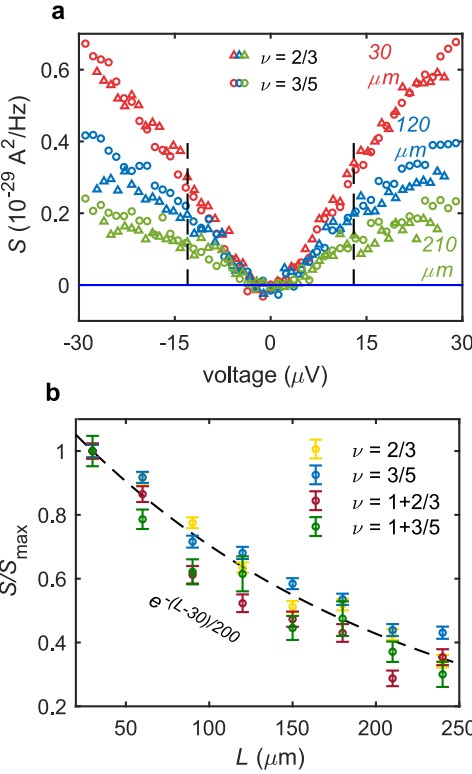

**Fig. 2 Length profile of the upstream noise in different states. a** Upstream noise as a function of the applied bias to the Source for $\nu = \frac{2}{3}$ (triangles) and $\nu = \frac{3}{5}$ (circles), for a few propagation lengths; 30 μm (red), 120 μm (blue), 210 μm (green). The dashed lines mark the voltage for which the length dependence profile was determined. **b** Length dependence of the upstream noise. The noise is normalized (with respect to shortest length) separately for each filling factor. Error bars represent the statistical error. The noise profile of all fillings matches nicely with exponential decay with a typical decay length of 200 μm (dashed line). This indicates the dominant role of dissipation rather than thermal equilibration.

flows downstream, recombination does not take place and current fluctuations are detected in contact A. Thus, the local temperature of the upstream modes $T_u(L)$ is encoded in the excess noise

$$S_{\text{excess}}^{\text{U}} \propto (T_u(L) - T_0), \tag{1}$$

where $T_0$ is the base temperature at A (we measured $T_0 =$ 11 mK at $\nu = 3/5$ and $T_0 =$ 14 mK at $\nu = 2/3$). The proportionality factor in Eq. (1) can depend on microscopic details of the edge modes, but importantly, it does not depend on $L$ (see Methods).

The measured noise is plotted in Fig. 2 at $\nu = \frac{2}{3}, \frac{3}{5}, 1 + \frac{2}{3}, 1 + \frac{3}{5}$, for $L$ in the range 30 − 210 μm. In Fig. 2a, the excess upstream noise is plotted as function of the voltage bias $V$ for three different $L$. Figure 2b displays the noise measured at the fixed voltage 13 μV, plotted as a function of $L$ for several states from distinct topological classes: at $\nu = \frac{3}{5}, \nu_Q < 0$; at $\nu = \frac{2}{3}, 1 + \frac{3}{5}, \nu_Q = 0$; and at $\nu = 1 + \frac{2}{3}, \nu_Q > 0$. Remarkably, we find that all noise profiles are similar. The noise strength vs $L$ fits nicely to a decaying exponent with a characteristic decay length of 200 μm. This suggests an unequilibrated regime: $l_{\text{eq}} > l_{\text{dis}} \approx 200$ μm.

**Two-terminal thermal conductance.** Next, we studied the thermal conductance, which supplements the upstream thermometry, as it is sensitive only to the heat returning to the source contact (and not to dissipation). We used two devices, B1 and B2, where a floating Ohmic contact was employed as the heat source (Fig. 3a). The floating contact, with area of a few tens of μm², was

connected to three separate arms of 2DEG. Simultaneously sourcing currents $+I$ and $-I$ from contacts $S_1$ and $S_2$, respectively, leads to a dissipation of power $P = I^2/G_{2T}$ in the Ohmic contact, while its potential remains zero. Upstream and downstream noise was measured (Fig. 3b) at contacts (depicted as amp DS and amp US) located in the upstream and downstream direction. Similar to device A, the length between the Ohmic contact and the upstream amplifier contact could be varied. Devices B1 and B2 were designed for short (15 − 85 μm) and long (35 − 315 μm) upstream distances $L$, respectively.

The downstream current fluctuations further allow extraction of the Ohmic contact temperature $T_m$ (the downstream noise, unlike the upstream noise, is independent of the local temperature of upstream modes at the downstream amplifier), which in turn determines the thermal conductance via a heat balance equation (see Methods). The normalized thermal conductance vs $L$ is plotted in Fig. 3c. Remarkably, the thermal conductance is length-independent for both $\nu = \frac{2}{3}$ and $\nu = \frac{3}{5}$ up to lengths of 315 μm. On the other hand, the upstream-noise decays by as much as 85% compared to the shortest length.

All measurements point to a lack of thermal equilibration, and instead to dissipation-dominated heat transport. As shown in Figs. 2b and 3c, the upstream-noise decays exponentially with $L$ independently of $\nu$. If the edges were equilibrated, one would expect distinct behavior of upstream noise depending on the topological class due to different heat transport characteristics. Hence, the upstream-noise data show that the thermal equilibration in the edge is not operative, and the noise decay is due to dissipation of energy to the environment. This is further supported by the length-independent thermal conductance at $\nu = \frac{2}{3}$, which is incompatible with diffusive transport in the thermally equilibrated regime. Note that dissipation does not affect the thermal conductance in our measurement scheme since the dissipated heat does not return back to the Ohmic contact[13].

**Quantitative analysis of the thermal conductance and upstream noise.** In the unequilibrated regime, the classical Johnson-Nyquist (JN) formula, used in similar experiments[10–12,26] should be corrected due to the mismatch between the upstream and downstream modes' temperature. The strength of the current fluctuations propagating downstream from a reservoir heated to a temperature $T_m$ is generally given by

$$S_{JN} = 2k_b G_{2T}(T_m - T_0)\alpha, \tag{2}$$

where $\alpha$ is a pre-factor that depends on microscopic details of the edge modes. The classical JN noise is restored ($\alpha = 1$) for a thermally equilibrated state with $n_d \geq n_u$, and in particular for any integer or particle-like fractional state (where $n_u = 0$). For unequilibrated $\nu = \frac{2}{3}$ and $\nu = \frac{3}{5}$ edges we find $\alpha = \frac{3}{4}$ and $\alpha = \frac{7}{10}$, respectively (see Methods).

At this point we can quantitively determine the thermal conductance. We plot the power $P$ dissipated at the Ohmic contact vs $T_m^2 - T_0^2$. A linear fit to the energy-balance equation $P = \frac{I^2}{G_{2T}} = 3\frac{\kappa_{2T}}{2}(T_m^2 - T_0^2)$ yields $\kappa_{2T}$ (see Methods). Note that $\kappa_{2T}$ determines the two-terminal thermal conductance of each individual arm, assuming that each arm contributes equally (which is the case in the absence of equilibration). For three integer states $\nu = 1, 2, 3$, the extracted thermal conductance agrees well with the expected values $\kappa_{2T}/\kappa_0 = n_d$ (see Supplementary Note 3). Here, the absence of upstream modes in these states makes thermal equilibration irrelevant. For the hole-conjugate states $\nu = \frac{2}{3}$ and $\nu = \frac{3}{5}$, we find $\kappa_{2T}/\kappa_0 = 1.00 \pm 0.03$ and $\kappa_{2T}/\kappa_0 = 1.45 \pm 0.03$, respectively (see Fig. 4a). For completeness, we point out that if one derives $T_m$ with the classical JN formula ($\alpha = 1$)[10–12,26], a significantly higher

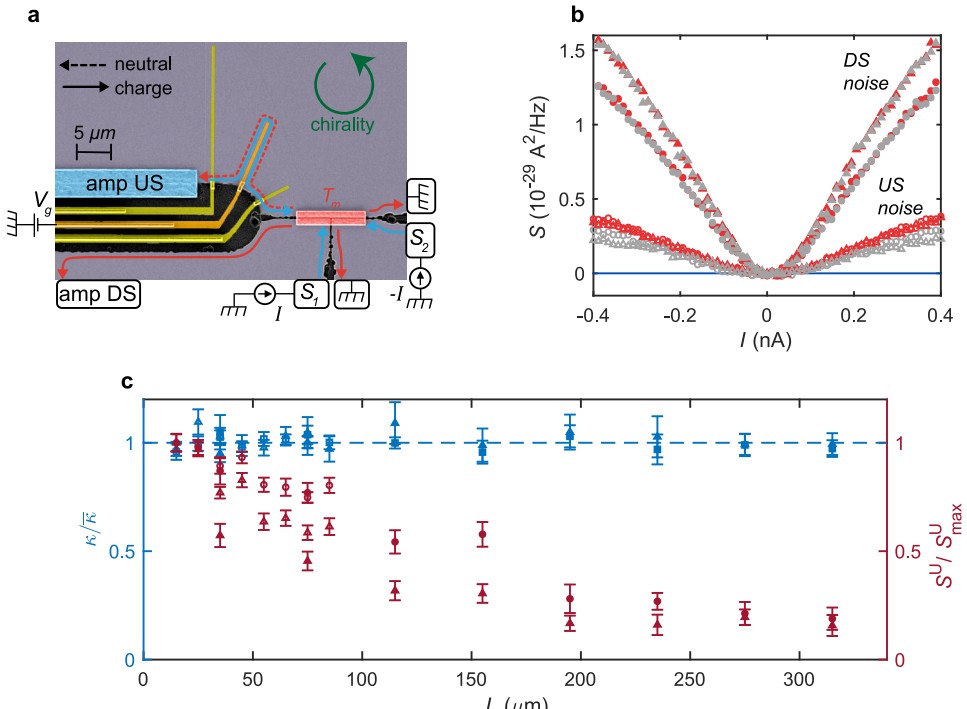

**Fig. 3 Length profile of the thermal conductance and the upstream noise. a** False colors SEM image of the central part of Device B1. The mesa (gray) is divided into three arms by the etched regions (black). The three arms are connected by a floating metallic island (in red) with area $15 \times 2\ \mu m^2$, serving as a heat source. When a current $I$ from $S_1$ and $-I$ from $S_2$ are sourced simultaneously, the floating island heats up to a temperature $T_m$. The resulting noise is measured simultaneously in the downstream and upstream amplifiers. The propagation length from the floating contact to the upstream amplifier can be varied using the metallic gates (yellow, as in Device A). Depicted is the case where the middle gate (darker yellow) redirects the path of the edge modes by the application of a gate voltage $V_g$, while the other gates are unbiased, and hence do not affect the propagation length. **b** Downstream noise (full shapes) and upstream noise (empty shapes) as a function of the current. Results are shown for $\nu = \frac{2}{3}$ (triangles) and $\nu = \frac{3}{5}$ (circles), and the propagation lengths 15 μm (red) and 75 μm (gray). The upstream-noise decays with length while the downstream noise does not. **c** Two-terminal thermal conductance $\kappa_{2T}$ (extracted from the downstream noise) (blue), and upstream-noise strength (red) as a function of length (See Methods). The thermal conductance is separately normalized for $\nu = \frac{2}{3}$ and $\nu = \frac{3}{5}$ with respect to their respective means. For both $\nu = \frac{2}{3}$ and $\nu = \frac{3}{5}$, we observe that $\kappa_{2T}$ is length-independent, while the upstream-noise decays (similarly to Fig. 2b). This indicates an unequilibrated thermal regime. The empty (full) shapes mark the data measured in device B1 (B2), and error bars represent the 95% confidence bounds of the normalized thermal conductance and the upstream-noise strength.

thermal conductance value is obtained: $\kappa_{2T}/\kappa_0 = 1.5$ and $\kappa_{2T}/\kappa_0 = 2.5$ for $\nu = \frac{2}{3}$ and $\nu = \frac{3}{5}$, respectively.

How can we understand the measured values of $\kappa_{2T}$? As shown in ref. [27] for $\nu = \frac{2}{3}$ (see Supplementary Note 11 for derivation and generalization to $\nu = \frac{3}{5}$), an emerging quantization of $\kappa_{2T}$ is expected as

$$\kappa_{2T}/\kappa_0 = (n_u + n_d) - 2\kappa_{12}/\kappa_0, \qquad (3)$$

in an intermediate transport regime $L_T \ll L \ll l_{eq}$, where $L_T$ is the thermal length. The first term in Eq. (3) is the expected value in the absence of the thermal equilibration, but it is lowered by the parameter $2\kappa_{12}/\kappa_0$ due to backscattering of plasmon modes at boundaries between contacts and the edge. Generally, $\kappa_{12}$ depends on interactions on the edge. When the system is close to a low-energy fixed point[28–30] at which a charge mode is decoupled from neutral modes, we find $\kappa_{12}/\kappa_0 = 2(1 - \nu)/(2 - \nu)$. Then, $\kappa_{2T}/\kappa_0 = 1$ for $\nu = 2/3$ with $n_u = 1, n_d = 1$, and $\kappa_{2T}/\kappa_0 = 13/7$ for $\nu = 3/5$ with $n_u = 2, n_d = 1$. For $\nu = 2/3$, our measured value agrees very well with the prediction[27] $\kappa_{2T}/\kappa_0 = 1$. For $\nu = 3/5$, the measured value is slightly lower than $13/7$, which might be related to a deviation of the system from the infrared fixed point. Interestingly, for $\nu = 2/3$, it was recently calculated that at the fixed point, the thermal equilibration length diverges[26].

We turn now to a quantitative analysis of the upstream noise. With vanishing thermal equilibration, we can write:

$$S_{excess}^{U} = 2k_B G_{2T} f_T (T_m - T_0) \equiv 2k_B G_{2T} f_T \Delta T, \qquad (4)$$

with $T_u(L)$ replaced by $T_m$ in comparison to Eq. (1). We denote the proportionality constant, $f_T$ as the thermal Fano factor. In Fig. 4b we plot the measured upstream excess noise as a function of $\Delta T$ for different lengths. All curves are linearly proportional to $\Delta T$, in agreement with Eq. (4). When $L$ decreases (and thus dissipation becomes less important), $f_T$ increases, but even at the shortest available length of 15 μm it is ~2 times smaller than predicted by our microscopic calculations (Supplementary Note 9). The reason for this discrepancy remains to be understood.

## Discussion

In summary, we demonstrated that counter-propagating modes efficiently exchange charge, but not energy. While we estimate the charge equilibration length to be shorter than 5 μm (see Methods), our observations set a lower bound on the thermal equilibration length $l_{eq} > l_{dis} \approx 200$ μm. These observations seem to agree (at least qualitatively) with recent measurements in short (edge length ≈ 5 μm) graphene samples[26], but disagree with previous measurements in GaAs[10]. We believe that the difference in equilibration efficiency between our finding and those reported in ref. [10] is due to microscopic differences between the measured devices. Such apparently important details include disorder, edge

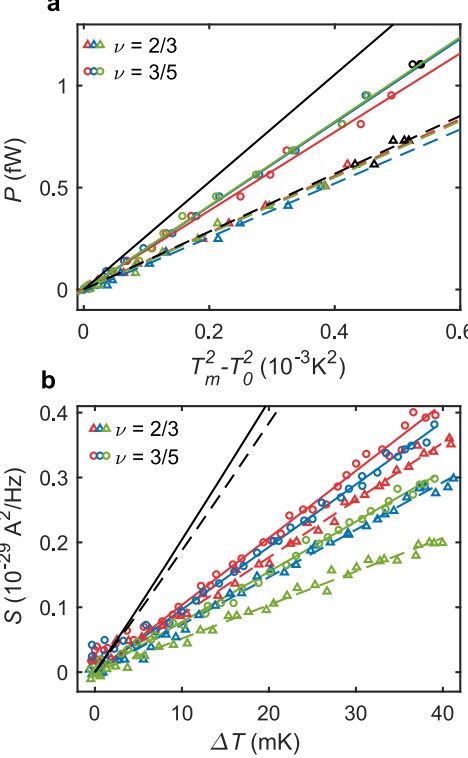

**Fig. 4 Quantitative analysis of the thermal conductance and the upstream noise. a** Dissipated power $P$ as a function of $T_m^2 - T_0^2$, where $T_m$ and $T_0$ are the Ohmic contact and the base temperatures, respectively. The base temperature was separately calibrated (see Supplementary Note 2) and found to be $T_0 = 11$ mK at $\nu = 3/5$ and $T_0 = 14$ mK at $\nu = 2/3$. The colored markers (low temperature data - $T_m < 25 mK$) were linearly fitted to extract $\kappa_{2T}$ (fits marked by colored dashed and full lines for $\nu = \frac{2}{3}$ and $\nu = \frac{3}{5}$, respectively). The black markers are high temperature points, which were not fitted, since at these temperatures, the cooling of the central contact by lattice phonons becomes non-negligible. We plot the data for $\nu = \frac{2}{3}$ and $\nu = \frac{3}{5}$ for propagation lengths 15 μm (red) 45 μm (blue), and 85 μm (green). We find length-independent, thermal conductances $\kappa_{2T}/\kappa_0 = 1.00 \pm 0.03$, $\kappa_{2T}/\kappa_0 = 1.45 \pm 0.03$ for $\nu = \frac{2}{3}$ and $\nu = \frac{3}{5}$, respectively. The theoretically expected values for $\nu = \frac{2}{3}$ ($\nu = \frac{3}{5}$) are plotted as a black dashed (full) line. We find excellent agreement with the data for $\nu = \frac{2}{3}$, while the thermal conductance for $\nu = \frac{3}{5}$ is somewhat smaller than predicted. **b** Excess upstream noise as a function of $T_m$ for $\nu = \frac{2}{3}$ and $\nu = \frac{3}{5}$, and for propagation lengths 15 μm (red), 45 μm (blue), and 85 μm (green). The slope of the linear fit, denoted as $2k_B G_{2T} f_T$ in Eq. (3), increases with decreasing length (due to diminishing dissipation) and approaches a value of roughly 0.5 times that predicted by a microscopic calculation. The predicted values (see Supplementary Note 9) are depicted by the black, dashed, and solid line for $\nu = \frac{2}{3}$ and $\nu = \frac{3}{5}$, respectively.

mode velocities, and inter-mode interaction. More research is required in order to fully understand the effect of such details. We also note that for co-propagating integer modes, the thermal equilibration has been reported to be much faster than charge equilibration[31]. Our observation of a thermally non-equilibrated transport regime for edges with counter-propagating modes provides important insights into the physics of hole-conjugate states.

## Methods

**Sample preparation**. The Ohmic contacts and gates were patterned using standard e-beam lithography-liftoff techniques. The Ohmic contact consists of Ni (7 nm), Au(200 nm), Ge(100 nm), Ni (75 nm), Au(150 nm) alloyed at 440°C for

50 seconds. To minimize strain-induced reflection from the gates, the surface was covered by 7 nm HfO$_2$ deposited at 200°C. After deposition, we etched the HfO$_2$, (except under the gates) using buffered oxide etch. The gate electrode consists of 5 nm Ti and 15 nm Au.

**Extraction of $T_m$ from downstream noise**. The excess downstream noise $S_{excess}^D$ in the three-arm devices B1 and B2 (Fig. 3a) is given by

$$S_{excess}^D = \frac{2}{3}\overline{(\Delta I_m)^2} + \frac{1}{9}\left(S_{excess}^{S_1} + S_{excess}^{S_2} + S_{excess}^U\right) - \frac{4}{3}G_{2T}k_B T_0, \quad \text{(M1)}$$

where the numerical factors come from the devices' geometry (see Supplementary Note 6 for details). Here, $\overline{(\Delta I_m)^2}$ is the noise from the current fluctuations emanating from the central floating contact and are thus related to the temperature $T_m$ of the central contact, while $S_{excess}^{S_1}$ and $S_{excess}^{S_2}$ are the excess noises generated at noise spots (Supplementary Fig. 6) near sources $S_1$ and $S_2$. Since $S_{excess}^{S_1}$ and $S_{excess}^{S_2}$ are analogous to the upstream noise, they can be taken into account by using measurements of $S_{excess}^U$ at upstream distances 30 μm (corresponding to the distance between $S_1$ and the central contact) and 150 μm (the distance between $S_2$ and the central contact), respectively.

The generated noise $\overline{(\Delta I_m)^2}$ on an edge segment is generally given by[21]

$$\overline{(\Delta I_m)^2} = \frac{2e^2}{h l_{eq}^C}\frac{\nu_-}{\nu_+}(\nu_+ - \nu_-)\int_0^L dx \Lambda(x)e^{-2x/l_{eq}^C} + \frac{2e^2}{h}k_B T_m\frac{(\nu_+ - \nu_-)^2}{\nu_+}, \quad \text{(M2)}$$

where $l_{eq}^C$ is the charge equilibration length and $\nu_+(\nu_-)$ is the total filling factor of the downstream (upstream) modes (e.g., $\nu_+ = 1$, $\nu_- = 1/3$ for $\nu = 2/3$ and $\nu_+ = 1$, $\nu_- = 2/5$ for $\nu = 3/5$). The noise $\overline{(\Delta I_m)^2}$ is the local noise generated by inter-channel electron tunneling, which is encoded in the noise kernel $\Lambda(x)$. Assuming absence of thermal equilibrium, $\Lambda(x)$ becomes independent of position $x$ [i.e., $\Lambda(x) = \Lambda_0(T_m, T_0)$], and thus Eq. (M2) is simplified as

$$\overline{\Delta I_m^2} = \frac{e^2}{h}\frac{\nu_-}{\nu_+}(\nu_+ - \nu_-)\Lambda_0(T_m, T_0) + \frac{2e^2}{h}k_B T_m\frac{(\nu_+ - \nu_-)^2}{\nu_+}. \quad \text{(M3)}$$

The simplified noise kernel $\Lambda_0(T_m, T_0)$ can be computed within a microscopic model (see Supplementary Note 7 for details). Specifically, we divide the edge segment into three regions: the left contact region, a central region, and the right contact region. The inter-channel interaction is taken to change sharply from zero in the contact regions to a finite value in the central region. The left and right contacts are taken at different temperatures $T_m$ and $T_0$, respectively. Within this model, we derive the formula $\Lambda_0(T_m, T_0) \simeq 2T_0 + 0.5(T_m - T_0)$, assuming strong interactions. The downstream excess noise $S_{excess}^D$ then reads

$$S_{excess}^D = \frac{1}{9}\left(S_{excess}^{S_1} + S_{excess}^{S_2} + S_{excess}^U\right) + \frac{4}{3}G_{2T}k_B(T_m - T_0)\frac{4\nu_+ - 3\nu_-}{4\nu_+}. \quad \text{(M4)}$$

Comparing to Eq. (2) and identifying $S_{JN} = \overline{(\Delta I_m)^2} - 2k_B G_{2T} T_0$, we find $\alpha = \frac{4\nu_+ - 3\nu_-}{4\nu_+}$. The central Ohmic contact temperature $T_m$ can be extracted from Eq. (M4). Ultimately, $T_m$ is used for the determination of both the thermal conductance $\kappa_{2T}$ and in plotting the upstream noise vs $\Delta T = T_m - T_0$.

**Extraction of $\kappa_{2T}$**. The thermal conductance $\kappa_{2T}$ is obtained from the heat balance equation

$$P = \frac{I^2}{G_{2T}} = \frac{3}{2}\kappa_{2T}(T_m^2 - T_0^2), \quad \text{(M5)}$$

which, in the steady state, equates the dissipated power and the emanating heat currents in the central contact. Here, we neglected all other mechanisms evacuating heat from the contact. The major correction to Eq, (M5) comes from the lattice phonons, which at low temperatures evacuate power proportional to $T_m^5$[32]. Radiative losses are completely negligible. In our device, the phonon contribution becomes important only at $T_m \sim 30 mK$. Thus we fitted our data only up to 25mK, where Eq. (M5) holds well. Moreover, we assume that all injected electrical power is dissipated in the central contact and raises its temperature (see Supplementary Note 10 for more details). The temperature $T_m$ in Eq. (M5) is extracted from downstream noise, Eq. (M4) as described above.

**Upstream-noise theory**. With the same model as for the downstream noise, the upstream excess noise is computed as

$$S_{excess}^U = \frac{3}{2}\frac{e^2}{h}\frac{\nu_-}{\nu_+}(\nu_+ - \nu_-)(T_m - T_0). \quad \text{(M6)}$$

The derivation of Eq. (M6) is given in the Supplementary Notes 7,9. In comparison to Eq. (4), we have $f_T = \frac{3}{4}\frac{\nu_-}{\nu_+}$. Eq. (M6) is used for the theoretical plots in Fig. 4b.

**Estimation of charge equilibration length**. A slightly unequilibrated charge conductance was reported at $\nu = \frac{2}{3}$ for very short edge distances[33]. In order to test

the equilibration of charge we sourced from $S_1$ AC voltage at the resonance frequency of the upstream amplifier (not DC current like the main measurements) and measured the voltage in the upstream amplifier. Given the sourced voltage $V_s$, we find by using the standard Landauer-Büttiker formalism,

$$V_{\text{amp}} = V_S \frac{G_U(L)}{3G_{2T}}, \tag{M7}$$

where $G_U(L)$ is the upstream conductance from the floating Ohmic contact to the upstream amplifier (the factor of 3 comes from the three arms of the B1 and B2 devices). In deriving Eq. (M7), we assumed $G_U \ll G_{2T}$. We see (Supplementary Fig. 5) that only at $\nu = \frac{2}{3}$ and $\nu = \frac{3}{5}$, and only when the propagation length is short, one would observe a finite $G_U$, which would indicate a not fully charge equilibrated edge. At the shortest available length $L = 15 \ \mu m$, we found $\frac{G_U}{G_{2T}} = 7 \times 10^{-3}$ for $\nu = \frac{2}{3}$ and $\frac{G_U}{G_{2T}} = 3 \times 10^{-4}$ for $\nu = \frac{3}{5}$. To rule out the possibility that the upstream current is a result of bulk currents due to finite longitudinal conductance, we repeated the measurement at a higher temperature. We observed that $G_U$ decreases at $21 mK$, as apparently the charge equilibration is faster. This behavior is inconsistent with bulk currents, since the longitudinal conductance is expected to increase with temperature. In a simple model for charge equilibration, we can write

$$G_U(L) = G_{U,0} e^{-\frac{L}{l_{\text{eq}}^C}}, \tag{M8}$$

where $G_{U,0} = \frac{e^2}{h} \nu_-$ is the zero length upstream conductance. From our data, we find $l_{\text{eq}}^C \approx 4 \ \mu m$ for $\nu = \frac{2}{3}$ and $l_{\text{eq}}^C \approx 2 \ \mu m$ for $\nu = \frac{3}{5}$. This charge equilibration length stands in sharp contrast to our observed thermal equilibration lengths, which are two orders of magnitude larger.

## Data availability

The datasets generated during and/or analyzed during the current study are available from the corresponding author on reasonable request.

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

## Acknowledgements

We thank M. Heiblum for his essential guidance and support throughout this project. We also thank Y. Gefen for many stimulating discussions in the course of this work. We acknowledge the help of Diana Mahalu with e-beam lithography. We acknowledge illuminating discussions with K. Snizhko. C.S. acknowledges funding from the Excellence Initiative Nano at Chalmers University of Technology. J.P. acknowledges funding by the Deutsche Forschungsgemeinschaft (DFG, German Research Foundation)—Projektnummer 277101999—TRR 183 (project A01). C.S. and A.D.M. acknowledge support by DFG Grants No. MI 658/10-1 and MI 658/10-2, and by the German-Israeli Foundation Grant No. I-1505-303.10/2019.

## Author contributions

R.A.M. and B.D. fabricated the devices and performed the measurements. R.A.M. analyzed the data with inputs from B.D., C.S., J.P., and A.D.M. C.S., J.P., and A.D.M. developed the theoretical model. V.U. grew the GaAs heterostructures. All authors contributed to the writing of the manuscript.

## Competing interests

The authors declare no competing interests.
