## [Peer Review File · Nature Communications]

REVIEWER COMMENTS

Reviewer #1 (Remarks to the Author):

The manuscript from Ron Aharon Melcer et al. reports on the experimental study with theoretical analysis of thermal equilibration on edges of different fractional quantum Hall (FQH) states. Four main experimental results were obtained and reported:

1. The upstream noise profiles are independent of the net chirality of edge modes (topological classes in Ref.[20]) in the observed FQH states. The measurement was performed at filling factors $\nu=2/3$, $\nu=3/5$, $\nu=5/3$, and $\nu=8/5$.
2. The two-terminal thermal Hall conductance is independent of the propagation length L in both FQH states at $\nu=2/3$ and $\nu=3/5$.
3. The upstream noise is significantly decreased in samples with long propagation lengths.
4. For both $\nu=2/3$ and $\nu=3/5$, the charge equilibration lengths were measured and estimated to be in the range of $2\mu\text{m}$ - $4\mu\text{m}$.

By combining (1)-(3), the authors concluded that the heat transport in the sample is dominated by dissipation with a very ineffective thermal equilibration between counterpropagating edge modes. The decay length was estimated to be $200\mu\text{m}$, whereas the thermal equilibration length is believed to be much longer. From (4), the authors further concluded that charge equilibration is more efficient than thermal equilibration in their sample. The conclusion of having a much longer thermal equilibration length than the charge equilibration length in counterpropagating edge modes is consistent with all previous studies. On the other hand, the lack of thermal equilibration in a setup with a propagation length longer than $200\mu\text{m}$ disagrees with a previous experiment performed also on GaAs heterostructures (Ref.[10]).

After reading the manuscript, I think the experiment was performed carefully and systematically. The work is remarkable in the sense that most of the experimental results agree nicely with the detailed theoretical analysis in the manuscript. The conclusions summarized before were reached in a consistent and logical way. Furthermore, I find the theoretical discussion of Johnson-Nyquist noise

in the unequilibrated regime, and the theoretical details in Secs.S7-S12 provide valuable resources to researchers working on a similar topic. Although the conclusion in the manuscript disagrees with Ref.[10], I do not think this disagreement deteriorates the present work. Instead, it directly reflects how subtle can be in interpreting data from thermal conductance experiment in quantum Hall systems and other topological states (such as spin liquid). In this respect, this work reminds researchers the importance of having a better understanding of thermal equilibration in quantum Hall edges. Therefore, the results in the manuscript are significant to the field. Based on the potential importance and good quality of the work, I would recommend a publication of the manuscript in Nature Communications.

Although I appreciate the work highly, there are several things in the manuscript which the authors should address or revise before publication. First, I find it rather difficult to find out the temperature at which the experiment was performed. For example, I can only find out the temperature of the Ohmic contact T_m from the caption of Fig.(4). I believe this is a very important piece of information, which the authors should state it (perhaps also T_0 , the base temperature) explicitly in the main text. Furthermore, the authors should point out clearly in the main text (instead of just discussing it in Sec.S10) the temperature range is low enough so that the phonon contribution in carrying out heat from the Ohmic contact is negligible. Also, the fitting of the experimental data has neglected high-temperature points. Is it because phonon contribution becomes significant in that temperature range?

Second, the authors have performed and reported data from upstream noise experiment at $\nu=2/3$, $\nu=3/5$, $\nu=5/3$, and $\nu=8/5$. However, I do not see any two-point thermal Hall conductance data at the last two filling factors. Did the authors perform the same type of measurement at that two filling factors? Due to the existence of the additional integer mode, I believe some interesting (yet possibly complicated) physics may happen there. For example, will $\kappa_{2T}/\kappa_0 \approx 2$ at $\nu=5/3$ due to partial equilibration between integer (the $\nu=1$ edge) and fractional modes (the $\nu=2/3$ edge) as predicted in Ref.[13]? If not, what is the corresponding expected value if thermal equilibration is completely absent? I believe such a discussion can provide important insights into future experiment, which can further test the absence or presence of thermal equilibration in more complicated quantum Hall edges. If the authors have performed the thermal Hall conductance measurement at $\nu=5/3$ and $\nu=8/5$, I suggest them to include and discuss the data as well.

Third, the authors compare the present result (lack of thermal equilibration) with a recent experiment performed in bilayer graphene (Ref.[25]). The thermal equilibration length at $\nu=2/3$ was predicted to be divergent at the strong-interaction fixed point ($\Delta=1$). If this divergent thermal equilibration length was indeed a correct feature in the present work, then the experimental results can be explained naturally. In the case of graphene devices, reaching the $\Delta=1$ fixed point may be facilitated by the atomically sharp confining potential. However (to the best of my knowledge), such a sharp confining potential is not commonly realized in GaAs heterostructures. Is the confining potential in the present setup much sharper than the corresponding potentials in Refs.[10, 11]? I

think the answer to this question may be an important key to resolve the disagreement between the current results and Ref.[10].

Finally, some other minor suggestions for the authors to consider are listed below.

1. In the second paragraph of the main text, the authors claimed that " ν_{Q} will be half-integers for non-Abelian states". While this statement is correct for FQH state at the filling factor $\nu=5/2$, I think this statement is somehow misleading. In general, ν_{Q} is the (net) central charge of the conformal field theory (CFT) describing the quantum Hall edge. Its value can be neither an integer nor a half-integer in some cases. A famous example is the Read-Rezayi state at level $k=3$, which its parafermion sector has a central charge $c=4/5$. This state is known to be a possible candidate for FQH state at $\nu=12/5$ although the nature of this FQH state is still unknown.

2. In the same paragraph, the authors cited Refs. [13-16] to emphasize the importance of understanding thermal equilibration on quantum Hall edges. For completeness, it is better to also include the recent work: H. Asasi and M. Mulligan, *Partial equilibration of anti-Pfaffian edge modes at $\nu=5/2$* , Phys. Rev. B **102**, 205104 (2020).

3. Fig.S6 is very messy. For example, the labels on the central contact overlap with each other. This makes it very difficult to read and understand the figure.

4. In Sec.S7B, the authors discuss the edge structure of FQH state at $\nu=3/5$. There, the authors stated that the edge consists of modes ϕ_1 , $\phi_{1/3}$, and $\phi_{1/15}$. Furthermore, they claimed ϕ_1 and $\phi_{1/3}$ are closer than ϕ_1 and $\phi_{1/15}$. I understand that this picture is quite obvious to experts in quantum Hall physics, but it will be much better if a suitable reference can be included there.

5. Some typos (which I can identify) should be fixed:

(a) On page 4, "at $\nu=3/5$, $\nu_{\text{Q}} > 0$...", it should be $\nu_{\text{Q}} < 0$.

(b) On page 5, "at $\nu=2/3$...", it should be at $\nu=2/3$.

(c) On page 5 of Supplementary Information, "microscopic properties...", it should be microscopic properties.

(d) On pages 8 and 9 of Supplementary Information, "the charge equilibration length is expected to increase with increasing temperature...", it should be decrease with increasing temperature.

Reviewer #2 (Remarks to the Author):

The manuscript reported the observation of fast equilibration of edge mode and inefficient equilibration of thermal signal. The finding has furthered the understanding of edge, and helped with the interpretation of previous half-integer thermal Hall conductance experiment. This work set an excellent example of the application of local thermometry technique.

In addition, this experiment creatively combines gate depleting and mesa etching. Due to the complexity of fabrication, measurements with different devices are not as convincing as one single device with different edge propagation lengths. If only the top gates are used without the etching, a thin and long gate sometimes is hard to keep connected, and the length could thus be shorter than expected. I think the authors' approach to creating different edge lengths in this work is neat.

I support the publication of this manuscript if the authors could address a few minor points.

1. The results in this manuscript are important enough and not directly related to the $5/2$ state. I don't see the need to mention the $5/2$ state in either the abstract or the conclusion. Of course, the physics of the $5/2$ state is extremely important and is probably the motivation of this work, but honestly this manuscript is not directly related to the $5/2$ state.

2. In Figure 1, if the symbols of US neutral and DS charge are of the same colors as plotted between contacts A and S, it may be easier for the readers to follow.

3. The description of noise spot is unclear in the main text. There is isolated information of noise spot in the supplementary information. How to realize a noise spot or how a noise spot is different from the rest of 2DGE might be important for the readers.

4. Last but not least, I have a technical concern. The temperature gradient is up to 40 mK and the power is in the order of fW. What's the background heat leak from the estimate of radiation and photo-electron interaction? The reliability of this measurement and the previous thermal conductance measurement is based on tiny power dissipation, so the order of magnitude heat leak estimate is crucial to judge the reliability of such a series of experiments.

Referee 1:

We thank the Referee for a very thorough reading of the manuscript, and for giving many insightful comments that help us improve our work. We are delighted to hear that Referee 1 thinks “the results in the manuscript are significant to the field” and, “I would recommend a publication of the manuscript in Nature Communications.”

Let us address the questions (marked in red) raised by the Referee:

First, I find it rather difficult to find out the temperature at which the experiment was performed. For example, I can only find out the temperature of the Ohmic contact T_m from the caption of Fig.(4). I believe this is a very important piece of information, which the authors should state it (perhaps also T_0 , the base temperature) explicitly in the main text. Furthermore, the authors should point out clearly in the main text (instead of just discussing it in Sec.S10) the temperature range is low enough so that the phonon contribution in carrying out heat from the Ohmic contact is negligible. Also, the fitting of the experimental data has neglected high-temperature points. Is it because phonon contribution becomes significant in that temperature range?

We thank the Referee for this important comment. We regret that this vital experimental information was not clearly visible in the original manuscript. In the revised manuscript, we have added the temperatures in which the experiments were performed ($T_0 = 11 - 14mK$) on page 4 below Eq. 1, and also in the caption of Fig.4. Furthermore, when $T_m, T_0 < 30 mK$ the phonon contribution (power proportional to T_m^5) is negligible, as evident by the linear dependence of the power with T_m^2 (this is consistent with data reported in Refs. [10,11]). The high-temperature data $T_m, T_0 > 25mK$ was thus neglected in our extraction of the thermal conductance. We have added a clarification on this issue in the caption of Fig.4 and also in the Methods section (page 9 of the revised manuscript below Eq. M5).

Second, the authors have performed and reported data from upstream noise experiment at $\nu=2/3$, $\nu=3/5$, $\nu=5/3$, and $\nu=8/5$. However, I do not see any two-point thermal Hall conductance data at the last two filling factors. Did the authors perform the same type of measurement at that two filling factors? Due to the existence of the additional integer mode, I believe some interesting (yet possibly complicated) physics may happen there. For example, will $\kappa_{-2T}/\kappa_0 \approx 2$ at $\nu=5/3$ due to partial equilibration between integer (the $\nu=1$ edge) and fractional modes (the $\nu=2/3$ edge) as predicted in Ref.[13]? If not, what is the corresponding expected value if thermal equilibration is completely absent? I believe such a discussion can provide important insights into future experiment, which can further test the absence or presence of thermal equilibration in more complicated quantum Hall edges. If the authors have performed the thermal Hall conductance measurement at $\nu=5/3$ and $\nu=8/5$, I suggest them to include and discuss the data as well.

We thank the Referee for this suggestion. Due to technical difficulties, we were not able to perform the two terminal thermal conductance measurements at $\nu = \frac{5}{3}$ and $\nu = \frac{8}{5}$. In fact, it is quite difficult to generate a good enough Ohmic contacts (especially in the low density material used) that generally work nicely for fragile states with an outer integer mode and inner fractional modes. In our device, the modes were not fully absorbed in the micron-sized contact (this is simple to detect by performing three-terminal conductance measurements). We call this

problem ‘initial reflection’. The initial reflection was found for $\nu = \frac{5}{3}$ and $\nu = \frac{8}{5}$ but not for $\nu = \frac{2}{3}$ and $\nu = \frac{3}{5}$. The presence of initial reflection compromises our measurement techniques for two main reasons. First, if the modes do not fully electrically couple to the contact, it is most likely that they are also not fully thermally coupled. Second, such a reflection is most likely associated with shot-noise. In our device we would not have been able to separate it from the thermal noise, thus preventing us from reliably measure the contact’s temperature.

Such a difficulty was not present in the device A as there the heat source is the hot spot generated at the back of a huge (hundreds of μm^2) Ohmic contact.

We agree with Referee 1 that such measurements, when performed carefully, are very important for the understanding of thermal equilibration. As Referee 1 points out, we also expect the thermal conductance $\kappa_{2T} \sim 2$ for $\nu = \frac{5}{3}$ under the assumption that the total charge mode is thermally decoupled with other modes (suggested in Ref. [13]). But, this type of assumptions for thermal equilibration should be tested in various filling factors in a careful and systematic way. This question is left for future research.

Third, the authors compare the present result (lack of thermal equilibration) with a recent experiment performed in bilayer graphene (Ref.[25]). The thermal equilibration length at $\nu=2/3$ was predicted to be divergent at the strong-interaction fixed point ($\Delta=1$). If this divergent thermal equilibration length was indeed a correct feature in the present work, then the experimental results can be explained naturally. In the case of graphene devices, reaching the $\Delta=1$ fixed point may be facilitated by the atomically sharp confining potential. However (to the best of my knowledge), such a sharp confining potential is not commonly realized in GaAs heterostructures. Is the confining potential in the present setup much sharper than the corresponding potentials in Refs.[10, 11]? I think the answer to this question may be an important key to resolve the disagreement between the current results and Ref.[10].

The Referee raises here a very important question. The simplistic answer is that apparently the edges in our GaAs sample could be sharp enough to drive the modes to the Kane-Fischer-Polchinski fixed point ($\Delta = 1$). So how can one explain the differences from reference [10], where the measurements were performed on GaAs samples processed using similar experimental techniques? The answer, most likely, involves difference in microscopic details between different samples. More specifically, the renormalization group flow of Δ is determined by the interplay between the edge velocities, disorder, and the inter-mode interaction. We note that different GaAs samples were reported to have very different edge velocities (although for integer modes) ranging from 0.55 to 4.3 [$10^5 m/s$] (see Table III in the SI in Ref. [31] in the revised manuscript). Given that Δ is close to one with increasing edge mode velocities, it would be good to measure the edge velocity and the thermal conductance in the same device and compare different samples. We have added a short discussion on this in the last paragraph (page 7).

Finally, some other minor suggestions for the authors to consider are listed below.

1. In the second paragraph of the main text, the authors claimed that " ν_Q will be half-integers for non-Abelian states". While this statement is correct for FQH state at the filling factor $\nu=5/2$, I think this statement is somehow misleading. In general, ν_Q is the (net) central charge of the conformal field theory (CFT) describing the quantum Hall edge. Its value can be neither an integer nor a half-integer in some cases. A famous example is the Read-Rezayi state at level $k=3$, which its parafermion sector has a central charge $c=4/5$. This state is known to be a possible candidate for FQH state at $\nu=12/5$ although the nature of this FQH state is still unknown.

We thank the Referee for attracting our attention to this issue. The Referee is perfectly correct, and we changed the phrasing in the main text to take this into account.

2. In the same paragraph, the authors cited Refs. [13-16] to emphasize the importance of understanding thermal equilibration on quantum Hall edges. For completeness, it is better to also include the recent work: H. Asasi and M. Mulligan, *Partial equilibration of anti-Pfaffian edge modes at $\nu=5/2$* , Phys. Rev. B **102**, 205104 (2020).

As suggested by Referee, we have added this reference in the revised manuscript.

3. Fig.S6 is very messy. For example, the labels on the central contact overlap with each other. This makes it very difficult to read and understand the figure.

We thank the Referee for pointing this out. Following the recommendation of the Referee, we have modified Fig S6 in the SI for increased visibility. In particular, we have clarified current fluctuations in different parts of the device by drawing "speech-bubbles".

4. In Sec.S7B, the authors discuss the edge structure of FQH state at $\nu=3/5$. There, the authors stated that the edge consists of modes φ_1 , $\varphi_{1/3}$, and $\varphi_{1/15}$. Furthermore, they claimed φ_1 and $\varphi_{1/3}$ are closer than φ_1 and $\varphi_{1/15}$. I understand that this picture is quite obvious to experts in quantum Hall physics, but it will be much better if a suitable reference can be included there.

We thank the Referee for suggesting how to address a wider audience for our work. As a reaction to this suggestion, we have in the revised SI clarified our assumptions on the structure of the $3/5$ edge. First, we have in the paragraph cited two papers discussing the $3/5$ edge structure: "Kane and Fisher, PRB 51 13449, (1995)" (Ref. S32 in the revised SI) and "Moore and Wen, PRB 57 10138, (1998)" (Ref. S31 in the revised SI). Secondly, we have included a simple and accessible motivation for the spatial separation of edge modes, based on the composite fermion picture. In this context, we added a reference to the seminal paper by Jain (PRL, 63 199, (1989)). It is listed as Ref. S33 in the revised SI. Finally, we have included also a reference to a recent paper: arXiv 2017.12616 (where two of us are co-authors). In this paper, we made the same simplifying assumptions for the $3/5$ edge as in the present manuscript, with excellent agreement between theory and experiment.

5. Some typos (which I can identify) should be fixed:

(a) On page 4, "at $\nu=3/5$, $\nu_Q > 0$...", it should be $\nu_Q < 0$.

(b) On page 5, "at $\nu=2/3$...", it should be at $\nu=2/3$.

(c) On page 5 of Supplementary Information, "microscopic properties...", it should be microscopic properties.

(d) On pages 8 and 9 of Supplementary Information, "the charge equilibration length is expected to increase with increasing temperature...", it should be decrease with increasing temperature.

We thank the Referee for spotting these typos. They have been corrected in the revised version.

=====

Referee 2:

We are grateful for the comments of the Referee for the thorough reading of the manuscript, and for insightful comments that helps us improve our work. We are happy to read that the Referee thinks "The finding has furthered the understanding of edge, and helped with the interpretation of previous half-integer thermal Hall conductance experiment", as well as the statement "I support the publication of this manuscript if the authors could address a few minor points."

Here we address these points:

1. The results in this manuscript are important enough and not directly related to the $5/2$ state. I don't see the need to mention the $5/2$ state in either the abstract or the conclusion. Of course, the physics of the $5/2$ state is extremely important and is probably the motivation of this work, but honestly this manuscript is not directly related to the $5/2$ state.

We appreciate the fact that the Referee found our results important regardless of a relation to the $\nu = \frac{5}{2}$ state. Indeed, the physics of abelian FQH edges is a very important issue by itself. At the same time, as was recognized by the Referee, understanding the $5/2$ state served as an additional strong motivation for the present work. Even though we didn't perform the measurement at $\nu = \frac{5}{2}$, we believe that our results are of primary importance for this field. The reason being that the strongest experimental evidence for the topological order of this state is the two terminal thermal conductance (reference 11). The interpretation of the results relies on the understanding of the extent of thermal equilibration between the counter propagating modes (as discussed by many theoretical papers, references 13 to 17 in the revised manuscript). A natural way to solve this problem of two unknowns (that is topological order and equilibration) is to directly measure the equilibration length for states with conceptually similar edge structure and a known ground state. We believe that our observation of macroscopically large equilibration length for $\nu = \frac{2}{3}$ and $\nu = \frac{3}{5}$ could have a strong influence on the ongoing debate regarding $\nu = \frac{5}{2}$. We thus feel that it is justified to keep the corresponding discussion in the introductory part of our article (pages 1-2) . At the same time, we follow the Referee's advice and remove mentioning of the $5/2$ state from the conclusion paragraph (page 7).

2. In Figure 1, if the symbols of US neutral and DS charge are of the same colors as plotted between contacts A and S, it may be easier for the readers to follow.

We thank the Referee for the suggestion to improve Fig. 1. In the revised manuscript, we have color-coded the legend with the same colors as in the device, for increased visibility.

3. The description of noise spot is unclear in the main text. There is isolated information of noise spot in the supplementary information. How to realize a noise spot or how a noise spot is different from the rest of 2DGE might be important for the readers.

The Referee raises here a very important question. We take the opportunity to emphasize that a noise spot is a property of any QH edge with counterpropagating, charge carrying channels under conditions where charge transport is equilibrated. The noise spots are regions of roughly the size of the charge equilibration length located on 'the downstream side' (the direction of electron motion in the magnetic field) of all attached contacts. Due to the chiral nature of the edge, only in these regions can inter-edge tunneling result in particle-hole pairs ending up in different contacts and generated low frequency noise. Inter-channel tunneling away from the noise spots lead, due to the equilibration, to particle-hole pairs that always end up in the same reservoir, and no noise is generated. Heating at the noise spot, e.g., by upstream heat transport, leads to increased noise.

This mechanism was first explained in Park et al PRB 99 161302 2019 (Ref. 25 in the revised manuscript) and generalized in Spanslatt et al PRL 123 137701 2019 (Ref. 21 in the revised manuscript).

To explain this mechanism in a clearer way, we have in the revised manuscript added a sentence at the end of page 3: "The existence of a noise spot is a consequence of counter-propagating edge modes and efficient charge equilibration.". We have also emphasized Refs. 21 and 25 once more.

4. Last but not least, I have a technical concern. The temperature gradient is up to 40 mK and the power is in the order of fW. What's the background heat leak from the estimate of radiation and photo-electron interaction? The reliability of this measurement and the previous thermal conductance measurement is based on tiny power dissipation, so the order of magnitude heat leak estimate is crucial to judge the reliability of such a series of experiments.

We thank the Referee for this comment. Due to the low temperatures in our experiment, we neglect the heat leaving the floating contact by any other mechanism other than by the QHE edge states. This approximation is supported by theory and has also been experimentally demonstrated (e.g. in Ref. 9 in the revised manuscript). Indeed, at higher temperatures, other heat evacuation mechanisms become active. We believe that the dominant one is the lattice phonons (dissipated power proportional to T_m^5) that becomes activated (in our device) around $30mK$. Thus, we only fit our two terminal thermal conductance data up to $25mK$, below which power is clearly proportional to T_m^2 and phonon cooling can be neglected. We have added a clarification on this issue to the caption of Fig.4 and to the Methods section (page 9 under Eq. M5).

Other mechanisms, such as cooling by radiation (say, black body for simplicity) are even more negligible. For temperatures $\sim 30mK$, and for a contact with surface area of few tens of μm^2 , the dissipated power from radiation (estimated by the Stephan Boltzmann law) is roughly 10^{-6} fW, i.e. 6 orders of magnitude less than that evacuated by the edge modes. Radiative corrections can thus be safely neglected.

REVIEWERS' COMMENTS

Reviewer #1 (Remarks to the Author):

The authors have successfully answered all my questions and provided necessary changes in the revised manuscript. Thus, I recommend this article to be published in Nature Communications.

Reviewer #2 (Remarks to the Author):

The authors have addressed all the comments from my previous referee report.